# Heatwave-Related Mortality Risk and the Risk-Based Definition of Heat Wave in South Korea: A Nationwide Time-Series Study for 2011–2017

**DOI:** 10.3390/ijerph17165720

**Published:** 2020-08-07

**Authors:** Cinoo Kang, Chaerin Park, Whanhee Lee, Nazife Pehlivan, Munjeong Choi, Jeongju Jang, Ho Kim

**Affiliations:** 1Department of Public Health Science, Graduate School of Public Health, Seoul National University, Seoul 08826, Korea; baton_onair@snu.ac.kr (C.K.); chaerin210@snu.ac.kr (C.P.); jleehwan33@gmail.com (W.L.); nazife@snu.ac.kr (N.P.); moon339@snu.ac.kr (M.C.); zzooo@snu.ac.kr (J.J.); 2Institute of Health and Environment, Seoul National University, Seoul 08826, Korea

**Keywords:** heatwave, mortality, nationwide study, risk-based heatwave definition, time-series analysis, urbanization

## Abstract

Studies on the pattern of heatwave mortality using nationwide data that include rural areas are limited. This study aimed to assess the risk of heatwave-related mortality and evaluate the health risk-based definition of heatwave. We collected data on daily temperature and mortality from 229 districts in South Korea in 2011–2017. District-specific heatwave-related mortality risks were calculated using a distributed lag model. The estimates were pooled in the total areas and for each urban and rural area using meta-regression. In the total areas, the threshold point of heatwave mortality risk was estimated at the 93rd percentile of temperature, and it was lower in urban areas than in rural areas (92nd percentile vs. 95th percentile). The maximum risk of heatwave-related mortality in the total area was 1.11 (95% CI: 1.01–1.22), and it was slightly greater in rural areas than in the urban areas (RR: 1.23, 95% CI: 0.99–1.53 vs. RR: 1.10, 95% CI: 1.01–1.20). The results differ by age- and cause-specific deaths. In conclusion, the patterns of heatwave-related mortality risk vary by area and sub-population in Korea. Thus, more target-specific heatwave definitions and action plans should be established according to different areas and populations.

## 1. Introduction

Heatwave is associated with increased mortality globally, with the elderly, women, and people with cardiovascular or respiratory diseases being more vulnerable to heatwave-related mortality [1,2,3,4,5,6,7,8]. The frequency, intensity, and duration of heatwave is projected to increase due to climate change, and the related mortality burden is accordingly anticipated to considerably increase in the near future [9,10]. Therefore, understanding the impact of heatwave on mortality is crucial for establishing and modifying action plans to mitigate the potential adverse effect of climate change on human health [11].

Previous studies have reported that, compared to residents in non-urban areas, the urban population is more vulnerable to heatwave, because they are more likely to be exposed to higher and more frequent heat due to the urban heat island phenomenon [5,12,13]. However, recent studies conducted in China showed a higher heat-related mortality risk in rural areas than in urban areas, and the lower levels of education, medical infrastructure, and socioeconomic status in rural areas might be associated with those results [14,15]. Despite these inconsistencies, few heatwave-mortality studies covering both urban and rural areas have been carried out, because it has been difficult to obtain weather data of rural areas due to the insufficiency or absence of a weather monitoring station [16]. Moreover, a nationwide study encompassing both urban and rural areas has not been conducted despite it having several benefits. A nationwide study can detect the spatial heterogeneity in heatwave exposures and excess heatwave-related mortality across all administrative areas. Furthermore, it can provide important scientific evidence to prioritize resource allocations and to support the policymaking for developing effective and impartial public health interventions under climate change [17].

To the best of our knowledge, there has been no nationwide study investigating the risk of heatwave on mortality and examining the risk-based definition of heatwave in South Korea (hereafter, Korea). The Korean Meteorological Administration releases a cautionary message, if the daily maximum temperature continues to be at 33 °C for at least 2 consecutive days, and a heat warning, if the daily maximum temperature continues to be at 35 °C for at least 2 adjacent days. However, previous studies have reported that the definition of heatwave may be substantially heterogeneous depending on regions in which climates, population, and environments are different [4,18]. In addition, a relative scaled heatwave definition based on the region-specific temperature distribution has been suggested to be more suitable for measuring the impact of the heatwave and detect the threshold point of heatwave-mortality risk that can be used as the region-specific cut-off point for raising a heatwave alarm [18,19,20]. Therefore, the definition of a heatwave, which can vary among regions in Korea, needs to be investigated on a national level. The corresponding results can be helpful to understand the differentiated heatwave definition for each region in Korea.

Accordingly, this study aimed to investigate the heatwave-mortality risk in Korea. Towards this goal, we used nationwide weather and mortality data from 229 districts collected over 7 years (2011–2017). We estimated the heatwave-mortality risk, investigated the health risk-based definition of heatwaves in the total areas, and examined whether the risk is different between urban and rural areas. In addition, we conducted sub-population analyses to assess differences in heatwave-risk patterns by sub-populations.

## 2. Materials and Methods

### 2.1. Study Area

All 229 second-level local authority districts (shi/gun/gu) in South Korea, which had a total population of 51,269,554 and a total area of 100,339.5 km^2^ in 2016, were included in the study.

### 2.2. Data Collection

We collected population-based daily time-series data on mortality and mean temperature for the 229 districts during 2011–2017. Because there were two areas merged or newly created to municipal areas, only two districts had different study years: Cheongju-shi (2014–2017) and Sejong-shi (2012–2017). Data for all-cause mortality and cause-specific mortality (cardiorespiratory and non-cardiorespiratory death) were collected from the Korea National Statistics Office. Data on temperature were provided by the real-time re-analysis data from the Korea Meteorological Office. The real-time data provided ‘dong’-level hourly temperatures across the whole country. The ‘dong’ is a smaller district level than the shi/gun/gu. ‘Dong’-level temperature data were calculated from the numerical weather prediction using a statistical modeling combination of the Regional Data Assimilation and Prediction System and Model Output Statistics, based on the Unified Model for all earth. We averaged 24-hour mean values across all dongs for each day in each shi/gun/gu district. The data were restricted to the summer season, identified as the four warmest months (June–September). Data sources are shown in Appendix A.

### 2.3. Definition of Heatwave

Relative cut-offs based on the district-specific daily mean temperature were used to define heatwaves. A series of heatwaves was defined as daily mean temperatures above certain percentiles (the 85th to 99th percentile, per a percentile) of the summer temperature distribution (June–September) for at least 2 adjacent days during the study period [21]. We used a duration of ≥2 consecutive days to define a heatwave, because previous studies showed that defining heatwaves using ≥2 consecutive days produced better statistical estimates than ≥3 and ≥4 days [21,22]. Furthermore, the duration effect of extreme temperature was less associated with increased mortality [22].

### 2.4. Urban–Rural Classification

We classified all local authority districts (shi/gun/gu) into urban and rural districts, based on the Local Autonomy Act of Korea [23], using the second-level administrative region in Korea. The first-level administrative region of Korea consists of seven metropolitan cities and nine provinces. The second-level administrative region is a district. The ‘gu’ district is defined as sub-districts of the seven metropolitan cities, whereas the ‘shi’ district is defined as sub-districts of nine provinces, which is an urbanized area with a population of more than 50,000, although there are some legislated exceptions. The ‘gun’ districts, which do not satisfy the criteria of ‘shi’ or ‘gu’, were classified as rural districts. Finally, all districts included in the study were classified as urban or rural areas, which included 147 and 82 districts, respectively.

### 2.5. Statistical Analysis

All statistical analyses were performed using R statistical software version 3.5.3 (R Foundation for Statistical Computing, Vienna, Austria, 2016)

#### 2.5.1. Two-Stage Time-Series Analyses

We used two-stage time-series analysis to derive district-specific and pooled heatwave-related mortality risk. In the first stage, we derived the lag-cumulative heatwave-related mortality risk for each district using a generalized linear model with quasi-Poisson distribution with a distributed lag model:$$Y_{t}\sim quasi-Poisson\left(\lambda_{t}\right)$$
(1)$$\log\left(\lambda_{t}\right)=\beta_{0}+s\left(x_{t};\eta \right)+\mathrm{factor}\left({\mathrm{DOW}}_{t}\right)+\mathrm{ns}\left({\mathrm{DOY}}_{t},df=4/\mathrm{years}\right)$$
where $Y_{t}$ is the observed death count on day *t*, $\lambda_{t}$ is the expected death count on day *t*, and $\beta_{0}$ is an intercept. As the primary exposures, binary indicators for a heatwave were included using a distributed lag model structure. We fitted the model using a flexible function of s(⋅): natural cubic spline with two interval knots at equally spaced on the log scale values with 10 lag days characterized by parameter η to capture the non-linear lagged effects of heatwave on mortality. To control for seasonality and overall temporal trends, a natural cubic B-spline of the day of the summer season within a year (ns(DOY_*t*_, *df* = 4/year)) with equally spaced knots and four degrees of freedom (*df*) was used. The day of week on day *t* (DOW_*t*_) was also controlled as a categorical variable. These modeling specifications were based on previous studies [24,25]. In the second stage, we pooled the estimates in the total population, for urban and rural areas, and for each sub-population using meta-analysis with potential confounder variables: population, local income tax, number of beds in the hospitals, proportion of total urban forest area, latitude, longitude, mean temperature, and range of daily mean temperature (Appendix A). All first- and second-stage analyses were repeated for each cut-off of heatwave (the 85th to 99th percentile of the temperature distribution).

#### 2.5.2. Identifying the Threshold Point of Heatwave

We applied a health risk-based definition to define the threshold point for heatwave. A piecewise regression (a V-shape regression) was used to estimate the threshold point.
(2)$$E\left(\log RR_{i}\right)=\beta_{0}+\beta_{1}\times PT_{i}+\beta_{2}\times \left(PT_{i}-\psi \right)$$
where log $RR_{i}$ is the log values of the pooled heatwave-related mortality risk in *i*th percentile of the temperature (85th to 99th percentiles of the daily mean temperature) and $\beta_{0}$ is an intercept. Explanatory variable $PT_{i}$ is all corresponding cut-off points (from 85–99) and ψ indicates threshold value. We performed the piecewise regressions by changing the cut-off values and identified the threshold points rendering the model with the lowest Akaike Information Criteria (AIC) [26]. The AIC rewards goodness of fit (based on the likelihood function) with a penalty function of the increasing estimated parameters. We selected the best threshold point from the model with the minimum AIC value. Threshold estimates were considered only if the slope above the estimated threshold point was positive (i.e., the threshold point was not defined in the inverse V-shape piecewise regression).

#### 2.5.3. Subgroup Analysis

As sub-population analyses, we repeated the aforementioned analyses for age (aged 65 y+ and aged 0–64 y) and cardiorespiratory (10th International Classification of Diseases [ICD-10]: I00–I99 or J00–J98)/non-cardiorespiratory deaths.

#### 2.5.4. Sensitivity Analysis

In the sensitivity analysis, we tested Equation (1) to examine whether our results are robust on showing difference patterns between urban and rural areas with various modeling conditions: (1) degrees of freedom for day of season (flexibility of seasonality and overall temporal trends; 2 to 4 *df*), (2) lag effect days (lagged responses between heatwave and mortality considering shorter and longer lag day effects: 7, 10, and 14 days), (3) degrees of freedom for lag (flexibility of lag response curve: 2 to 3 *df*), and (4) heatwave duration (consecutive days exposed to heatwave: ≥2 days and ≥3 days). All-cause death was used in the sensitivity analysis.

## 3. Results

A total of 590,639 deaths were observed during the study period; of these, 493,911 were in urban areas, and 96,728 were in rural areas. Figure 1 shows the geographical distribution of urban and rural areas (Figure 1A) and the average summer temperature for each of the 229 districts (Figure 1B). The average summer temperatures were generally higher in urban areas than in rural areas (Figure 1B). Appendix A displays the geographical location of South Korea.

Table 1 summarizes the descriptive statistics for mortality and temperature variables. The mean average summer temperature in urban areas was higher than that in rural areas (23.7 vs. 22.7 °C). The average count of all-cause death was 2579; of these, 1867 were among those aged 65 y+, whereas 712 were among those aged 0–64 y. There were 758 and 1821 average cardiorespiratory and non-cardiorespiratory deaths, respectively. The death counts were generally greater in urban areas than in rural areas (3360 vs. 1180); however, the average number of deaths per 100,000 people was greater in rural areas (2363) than in urban areas (1214).

Figure 2 shows the heatwave-related mortality risks by each cut-off of the heatwave definition. In the total areas, the heatwave-related mortality risk generally increased as the cut-off value increased. However, the increasing pattern differed between the urban and rural areas. In the urban areas, the heatwave-related mortality risk increased gradually without a marked increasing point. In contrast, the heatwave risks were nearly a flat shape before a 95th percentile cut-off point in rural areas; after that point, the risks escalated rapidly. The maximum heatwave-related mortality risk was estimated on the 99th percentile of the average temperatures in all areas.

The results on the threshold point of heatwave-mortality risk and risk-maximum points with corresponding heatwave-related mortality risks are shown in Table 2. The threshold point for the total area was estimated at 93%, and the estimated threshold points were lower in urban areas than in rural areas (92% vs. 95%). In addition, the heatwave-related mortality risks at the risk-maximum points were greater in rural areas than those in urban areas (RR: 1.23, 95% CI: 0.99–1.53 vs. RR: 1.10, 95% CI: 1.01–1.20, respectively; Wald-type *p*-value: 0.013).

Figure 3 shows the heatwave-related mortality risk according to each cut-off of the heatwave definition by age groups and cause of death. In the overall population, we found higher heatwave-related mortality risks for those of the people aged 65 y+ compared to people than in those aged 0–64 y. Meanwhile, the people aged 65 y+ living in urban areas showed a declining heatwave risk pattern after the 95th percentile cut-off point. Such pattern was not shown in the rural areas. We also found that the heatwave-related mortality risk pattern was different between cardiorespiratory-related deaths and non-cardiorespiratory deaths. In general, the heatwave-related mortality risk for non-cardiovascular mortality was higher than that for cardiovascular mortality in urban areas. An opposite pattern was observed in rural areas. In addition, the heatwave-related cardiorespiratory mortality risk in urban areas decreased after the 95th percentile cut-off.

Among the people aged 65 y+ living in urban areas, there was no threshold point of heatwave, and the maximum heatwave-related mortality risk was 1.13 (95% CI: 1.08–1.18) at the 95th percentile cut-off (Table 3). Meanwhile, in rural areas, the threshold point was at the 93rd percentile of the cut-off temperature, and the maximum heatwave-related mortality risk was 1.20 (95% CI: 1.02–1.40) at the 98th cut-off for those people aged 65 y+. Meanwhile, in urban areas, the maximum heatwave-related risk for non-cardiorespiratory mortality (RR: 1.09, 95% CI: 1.04–1.13) was higher than that for cardiorepiratory mortality. While, in rural areas, the maximum heatwave-related risk for cardiorespiratory mortality (RR: 1.32, 95% CI: 1.02–1.71) was greater than that for non-cardiorespiratory mortality.

Sensitivity analysis showed that our results were robust (showing similar pattern according to the threshold points) even after changing the model choices. Changing the degrees of freedom for the day of the season (2–4 *df*), lag days (7, 10, and 14 days), degrees of freedom for lag (2–3 *df*), and heatwave duration (≥2 days, ≥3 days) showed similar results (Appendix A).

## 4. Discussion

The present study estimated the heatwave-related mortality risk and examined the risk-based threshold point of heatwave in the total, urban, and rural areas in Korea using a district-level nationwide time-series data covering all 229 administrative districts. In the total areas, the threshold point was determined at the 93rd percentile of daily mean temperature distribution, and the maximum risk of heatwave was 1.11 (95% CI: 1.01–1.22). The threshold point was lower in urban areas than that in rural areas (92nd vs. 95th percentiles). Meanwhile, the maximum heatwave-related risk was higher in rural areas than that in urban areas. These findings were generally more evident in people aged 65 y+ than in those aged 0–64 y. The heatwave-related risk of non-cardiorespiratory mortality was higher than the risk of cardiorespiratory mortality in urban areas, whereas the opposite trend was observed in rural areas.

The threshold points and the maximum risk point for heatwaves were observed around the 95th percentile and the 99th percentile of the mean temperature, respectively. These findings are consistent with those of previous studies [22,25,27]. Guo et al. evaluated 400 communities in 18 countries and reported that higher temperature cut-offs had higher effect estimates for the heatwave in almost every country [22]. A study in Korea and Japan also reported that the heatwave-related mortality risk sharply increased at the 95th percentile of daily mean temperature and showed the maximum heatwave-mortality risk at the 99th percentile [25]. Another study reported that heatwave-related mortality relative risks started to increase around the 95th percentile of the temperature, and the highest heatwave-related risk was identified at the 99th percentile in Melbourne, Australia [27].

In addition, our study showed that the threshold point was lower in urban areas than in rural areas; we hypothesized that this might be related to the heat island phenomenon in urban areas [12,13,28]. The urban heat island effect, which occurs when a huge amount of heat generated by urban structures re-radiates solar radiation, is known to increase temperatures in urban areas [29]. As shown in Figure 1B, the urban heat island phenomenon was also observed in urban areas in Korea. Furthermore, we found that the absolute-scaled temperatures (Celsius) corresponding to the threshold point were analogous between urban and rural areas. In addition, in the population aged 65 y+, an increasing trend of the heatwave-related mortality risk in the low cut-off values (85th to 95th percentiles) was more prominent in urban areas than that in rural areas (Figure 3). These results imply the necessity of an early heatwave warning system during moderately high summer temperatures in urban areas.

Although we found a greater maximum heatwave-related mortality risk in rural areas than in urban areas, a rapid increase in heatwave-related mortality risk above the threshold point was observed in rural areas. This pattern was different from urban areas showing constant or attenuated risk patterns above the threshold. Previous studies have reported that urban areas generally have a higher risk of heatwave-related mortality [12,28,30], and the urban heat island phenomenon has been suggested as the major factor of the higher risk [31]. However, several studies showed that the heat-related mortality risk was higher in rural areas than in urban areas [15,32] and suggested the lower levels of non-climatic factors (e.g., air conditioner prevalence, education, medical infrastructure, and aging population in rural areas) were associated with a higher heat risk in rural areas [14,15]. We also postulated that the higher heatwave risk in the Korean rural areas could be partly explained by the demographic and socioeconomic characteristics and medical infrastructure of rural areas. Specifically, the percentage of agriculture, forestry, fisheries industries [33], and the elderly population are generally higher in rural areas than that in urban areas (Appendix A). In addition, the lower availability of hospital beds (Appendix A) indicates a lower level of medical infrastructure in rural areas than in urban areas. We surmise that the aforementioned occupational and demographic characteristics in rural areas may be associated with the rapid increase in heatwave-related mortality risk above the threshold point. Because the economic activities in rural areas are generally conducted outdoors in the summer, the impact of a heatwave during a period of extreme heat can be amplified. Meanwhile, the higher levels of air conditioner usage and internet/mobile network in urban areas, which can contribute to higher accessibility to heatwave alarms and dissemination of relevant health information, could be a factor in the lower heatwave risk in urban areas as compared with rural areas. However, our study is limited in providing evidence for statistical estimates; therefore, future research should verify the related factors that are hypothesized to explain the differential heatwave risk between urban and rural areas.

Our findings also showed that the heatwave-related risk of non-cardiorespiratory mortality was greater than that of cardiorespiratory mortality in urban areas. However, rural areas showed the opposite pattern. We postulated that the results may be related to better and more advanced healthcare systems (i.e., medication and management system) for patients with cardiorespiratory diseases in urban areas. A study has reported that healthcare advances have decreased the cardiorespiratory disease-related mortality rates in the recent decades [34], and we also observed more advanced healthcare services in urban areas than in rural areas, as evidenced by the higher average number of clinics and physicians per 100,000 population in urban areas than that in rural areas (Appendix A). However, we could not identify any association between an advanced healthcare system and lower heatwave-related mortality risk, and thus, this should be investigated in future studies.

The study findings can suggest further implications about the current heatwave warning system implemented in Korea. Currently, a heatwave watch/warning is released when daily maximum temperatures above 33/35 °C are predicted to last for more than two consecutive days; these criteria are identical across all regions in Korea. From our data, we found that a daily maximum of 33 °C corresponded to about the 96th percentile of the daily mean temperature in all areas. A daily maximum of 35 °C corresponded to about the 98th percentile of the daily mean temperatures in all areas (Appendix A). These percentiles were generally higher than the risk based threshold we examined. Our results also show that the threshold points of the heatwave are heterogeneous between urban and rural areas, indicating the need for area-specific definitions and alarm policies for heatwave. In addition, our results indicate that the patterns of heatwave-related mortality risk differ by age and causes of death, and thus target-specific action policies against heatwaves should be considered to reduce the adverse impacts of heatwave on health.

This study has some limitations. Previous studies have reported that the vulnerability to heat changed during the last decades [25,35] and accordingly suggested that the definition of heatwave should be changed over time [25]. However, we could not consider the temporal variations in heatwave-related mortality risk, due to our short study period (2011–2017). This further examination should be performed with a longer study period. Second, our study is limited in investigating the potential confounding role of relative humidity and air pollutions in the heatwave-related mortality risk due to data limitation, and thus we could not eliminate the possibility of bias in our results. However, several studies showed that relative humidity and air pollutions have a relatively smaller impact on mortality than temperature [36,37]. Further, it has been argued that these variables should not be considered as confounders of the temperature-mortality relationship [37,38]. Finally, this study used the second-level administrative district data; therefore, our sample size was not fully sufficient to calculate the reliable confidence intervals of estimates in the rural areas and cause of death-specific analyses.

Despite these limitations, this study remains valuable, because to the best of our knowledge, this is the largest study to examine the heatwave-related mortality risk and risk-based definition of heatwave in Korea using nationwide data. Particularly, this study included rural districts that have been rarely studied previously. In addition, we used the district-level time-series data that has higher spatial resolution than metropolitan city-level data used in previous Korean studies [25,26].

## 5. Conclusions

The present study investigated the heatwave-related mortality risk and risk-based definition of heatwave in Korea using a nationwide data. The patterns of heatwave-related mortality differed between urban and rural areas, by age group (people aged 0–64 y and 65 y+) and cause of death (cardiorespiratory and non-cardiorespiratory). Furthermore, we evaluated age- and cause of death-specific patterns of the heatwave-mortality risk, and our results can be used to establish a more effective community-level heatwave response system. Moreover, our findings can contribute to developing the personalized heatwave warning system and action plans in Korea for people who are more vulnerable to heat.

## Figures and Tables

**Figure 1 ijerph-17-05720-f001:**
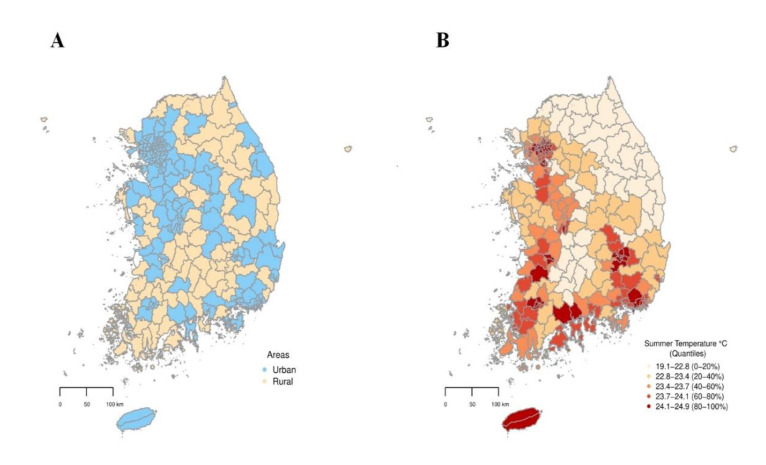
Geographical distributions of urban and rural districts (**A**) and average summer temperatures (**B**) in South Korea during 2011–2017.

**Figure 2 ijerph-17-05720-f002:**
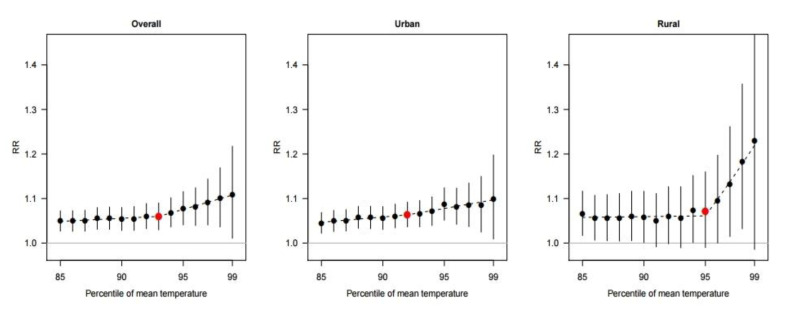
Associations between the percentile of the mean temperature and pooled heatwave-related mortality risk (RR) in the total, urban, and rural areas with 95% confidence interval (vertical lines). The dashed line indicates the relative risk based on the piecewise regression (a V-shape regression) model. The red points indicate the estimates at threshold points.

**Figure 3 ijerph-17-05720-f003:**
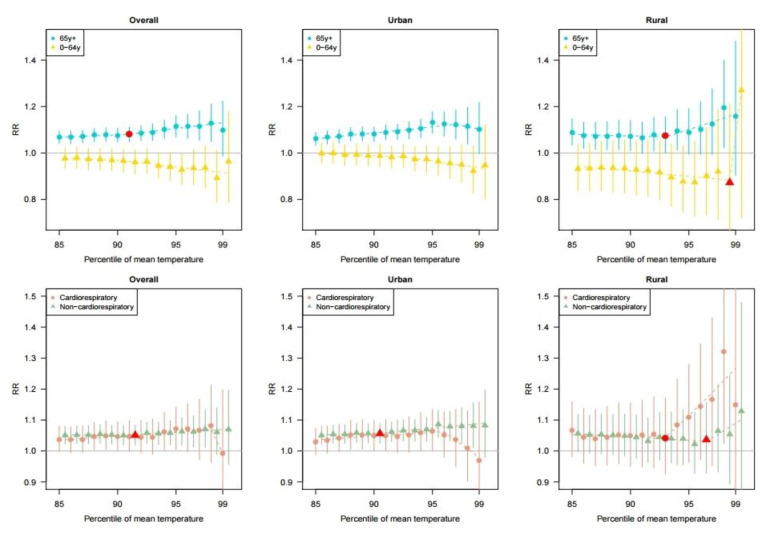
Age-specific and mortality cause-specific associations between the percentile of the mean temperature and pooled heatwave-related mortality risk (RR) in the total, urban, and rural areas with 95% confidence interval (vertical lines). The dashed line indicates the relative risk based on the piecewise regression (a V-shape regression) model. The red points indicate the estimates at the threshold points. The threshold points were excluded if the slope after the threshold point was negative.

**Table 1 ijerph-17-05720-t001:** Descriptive statistics of the mean temperature and daily mortality in the total, urban, and rural areas during the summer of 2011–2017. Data are averages of mean temperature (range), population (range), death counts (range), and deaths per 100,000 people (range).

Mean (Range)			
	Total (229 Districts)	Urban (147 Districts)	Rural (82 Districts)
**Mean temperature (°C)**	23.3 (19.1, 24.9)	23.7 (19.1, 24.9)	22.7 (20.0, 24.3)
**Population (2011–2016)**	223793.6 (10392.8, 1151632.8)	318843.4 (41727.5, 1151632.8)	53399.4 (10392.8, 212373.7)
Deaths			
Total	2579.2 (148, 10929)	3359.9 (344, 10929)	1179.6 (148, 2561)
Aged 65 y+	1867.2 (107, 7493)	2379.6 (250, 7493)	948.6 (107, 2013)
Aged 0–64 y	711.6 (41, 3436)	979.9 (94, 3436)	230.7 (41, 745)
Cardiorespiratory	758.4 (31, 3432)	970.1 (91, 3432)	378.9 (31, 813)
Non-cardiorespiratory	1820.8 (117, 7497)	2389.8 (253, 7497)	800.7 (117, 1748)
**Deaths per 100,000 persons**			
Total	1625.2 (612.9, 3370.1)	1213.6 (612.9, 2458.0)	2363.0 (1177.5, 3370.1)
Aged 65 y+	1251.2 (433.9, 2871.0)	878.9 (433.9, 2043.3)	1918.6 (837.9, 2871.0)
Aged 0–64 y	373.7 (176.9, 537.2)	334.5 (176.9, 517.7)	443.9 (278.9, 537.2)
Cardiorespiratory	504.1 (171.4, 1273.3)	361.7 (171.4, 834.0)	759.3 (298.3, 1273.3)
Non-cardiorespiratory	1121.1 (435.8, 2413.1)	851.9 (435.8, 1687.6)	1603.7 (817.5, 2413.1)

**Table 2 ijerph-17-05720-t002:** Threshold point, cut-off point with the maximum RR, and the maximum value of pooled heatwave-related mortality risk (RR) in the total, urban, and rural areas.

	Total (229 Districts)	Urban (147 Districts)	Rural (82 Districts)
Threshold point (% of temperature)	93%	92%	95%
Maximum heat-RR (95% CI)	1.11 (1.01, 1.22)	1.10 (1.01, 1.20)	1.23 (0.99, 1.53)
Cut-off point with the maximum RR (%)	99%	99%	99%

**Table 3 ijerph-17-05720-t003:** Sub-population analysis results of the threshold point, cut-off point with the maximum RR, and the maximum value of pooled heatwave-related mortality risk (RR) by age and mortality cause group in the total, urban, and rural areas.

		Total	Urban	Rural
		(229 Districts)	(147 Districts)	(82 Districts)
Aged 65 y+	Threshold point (% of temperature)	91%	- ^1^	93%
	Maximum heat-RR (95% CI)	1.13 (1.05, 1.21)	1.13 (1.08, 1.18)	1.20 (1.02, 1.40)
	Cut-off point with the maximum RR (%)	98%	95%	98%
Aged 0–64 y	Threshold point (% of temperature)	- ^1^	- ^1^	98%
	Maximum heat-RR (95% CI)	0.98 (0.93, 1.03)	1.00 (0.96, 1.04)	1.27 (0.72, 2.24)
	Cut-off point with the maximum RR (%)	86%	86%	99%
Cardiorespiratory	Threshold point (% of temperature)	- ^1^	- ^1^	93%
	Maximum heat-RR (95% CI)	1.08 (0.96, 1.21)	1.06 (1.00, 1.13)	1.32 (1.02, 1.71)
	Cut-off point with the maximum RR (%)	98%	95%	98%
Non-cardiorespiratory	Threshold point (% of temperature)	91%	90%	96%
	Maximum heat-RR (95% CI)	1.07 (1.01, 1.13)	1.09 (1.04, 1.13)	1.13 (0.86, 1.48)
	Cut-off point with the maximum RR (%)	97%	95%	99%

^1^ The threshold points were not estimated because the slope after the predicted threshold point using a piecewise regression (a V-shape regression) model was negative.

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
