# Peer review of "Heatwave-Related Mortality Risk and the Risk-Based Definition of Heat Wave in South Korea: A Nationwide Time-Series Study for 2011–2017"

_ijerph, 2020, doi:10.3390/ijerph17165720_

Round 1
Reviewer 1 Report
This interesting study has raised a good point that a national heatwave definition might not be suitable for every city. The analytical approach used seemed sound, and the way the authors presented the findings was decent. Three concerns were raised when I read the manuscript.
1. The authors did not seem to have mentioned if they had or how they controlled for long-term trend in the data analysis;
2. Since 33 °C and 35 °C are the two important thresholds for the national heat early warning of South Korea, it would be interesting if the authors could clarify where these two thresholds sit between 85th percentile to 99th percentile;
3. It has been increasingly reported in recent years that diabetics are also very vulnerable to heat impact. If the burden of diabetes in South Korea is high, it might be worth it to include diabetes in the stratified analysis as well.
Author Response
file attached

Reviewer 2 Report
1 Temperature-humidity index (THI) is usually regarded as heat stress response of the animals and human, why not use THI and its risk of human disease? The author only focus on the temperature, it is insufficient, the author should incorporate effects of humidity and temperature (THI) in this study.
2 Heat wave induced human death due to heat stroke. The author divided the factors into Cardiorespiratory, no-Cardiorespiratory, and age, please provide the correlations with stroke. How do you exclude the deaths unrelated to heat stroke?
Heat stroke is an acute disease characterized by central nervous and/or cardiovascular dysfunction caused by thermoregulatory central dysfunction, failure of sweat glands, and excessive loss of water and electrolytes in the heat season, high temperature, and/or high humidity. Please supplement the list of human diseases, analyze the relationship between detailed disease and heat wave, it will be beneficial to the society.
Author Response
file attached

Reviewer 3 Report
The paper is valuable as a statistical study of heat-related mortality with respect to urban-rural difference over South Korea. The following are some comments for consideration.
[Main comments]
(1) It will be better to describe the procedure of statistical analysis (Section 2.5) in more detail, in order to make it understandable without ambiguity, even if it is written in references. It is recommended to use equations to indicate some basic concepts such as the "quasi-Poisson distribution with a distributed lag model" (Line 110), "natural cubic spline including an intercept with two interval knots at equally spaced log values of 10 lag days" (Line 112), and "piecewise regression (a V-shape regression)" (Line 125). It is also recommended to present the definition of "threshold point", which is one of the main theme of the study.
Additionally, it is recommended to write the meaning of concepts related to the sensitivity analysis, such as the "day of season", "lag effect days", "degrees of freedom for lag", and "heatwave duration". Please explain how they are related to the degree of freedom.
(2) I have a concern that the confidence ranges of the results are rather large as shown in figures and tables, so that the reliability of detailed features obtained in the analysis may be limited. It will be better to acknowledge this matter as a limitation of the study.
[Other comments]
@ Please write the exact definition of percentiles (Line 89). Are they defined from daily values throughout the year, or from values for June to September?
@ Line 149 "The average number of deaths was generally greater in urban areas than that in rural areas." --- It will be more meaningful to discuss the number of deaths relative to population.
@ Please write the meaning of vertical bars in Figs.2 and 3.
@ Line 191 "In the urban areas, the threshold point of heatwave was estimated at the 91st percentile of temperature cut-off in urban areas" --- I think this sentence is for people of >=65 years old.
@ Line 194 "the threshold point was at the 93rd percentile of the cut-off temperature" --- I think this sentence is for people of >=65 years old.
@ Line 265 "the heatwave-related risk of cardiorespiratory mortality was ---" --- Do you mean non-cardiorespiratory mortality?
Author Response
file attached

Reviewer 4 Report
REVIEW - "Heatwave-related mortality risk and the risk-based definition of heat wave in South Korea: A nationwide time-series study for 2011–2017"
The article called "Heatwave-related mortality risk and the risk-based definition of heat wave in South Korea: A nationwide time-series study for 2011–2017" is well structured with a robust methodology. This research is relevant to Health Climatology and Geography areas and to South Korea indeed. There are few suggestions to collaborate with this good material:
Line 129: what is the Akaike Information Criteria? It would be appreciable if the authors could explain this criteria to the reader.
Figure 1: Where are the geographical coordinates of South Korea? Where are the geographical location of South Korea in Asia? These are important informations for a non-South Korean reader.
Table 1: I am observing the total-urban-rural data columns. Are the number of deaths counted right? Are the death numbers about (aged 65 y+) and (aged 0-64y) being inserted at the wrong lines? Please check the table carefully.
Thank you.
Author Response
file attached

Round 2
Reviewer 2 Report
The author has responsed to the comments.
Reviewer 3 Report
I appreciate the authors' effort of revision. The article is ready for acceptance.
As a minor comment, the expression "to examine our results are robust on showing" in Line 171 is grammatically awkward. It should be "to examine the robustness of our results by ---" or something.